# Laboratory data on wave propagation through vegetation with following and opposing currents

Zhan Hu[1,2,3], Simei Lian[1,3], Huayu Wei[1,4], Yulong Li[5,*], Marcel Stive[6], Tomohiro Suzuki[6,7]

[1]School of Marine Science, Sun Yat-Sen University, and Southern Marine Science and Engineering Guangdong Laboratory (Zhuhai), Zhuhai, 519082, China
[2]Guangdong Provincial Key Laboratory of Marine Resources and Coastal Engineering, Guangzhou, 510275, China
[3]Pearl River Estuary Marine Ecosystem Research Station, Ministry of Education, Zhuhai, 519082, China
[4]Department of Ocean Science, Hong Kong University of Science and Technology, Hong Kong, China
[5]Technology Centre for Offshore and Marine, 119077, Singapore
[6]Faculty of Civil Engineering and Geosciences, Delft University of Technology, Stevinweg 1, Delft 2628 CN, the Netherlands
[7]Flanders Hydraulics Research, Berchemlei 115, Antwerp 2140, Belgium

*Correspondence to*: Yulong Li (li_yulong@tcoms.sg)

**Abstract.** Coastal vegetation has been increasingly recognized as an effective buffer against wind waves. Recent laboratory studies have considered realistic vegetation traits and hydrodynamic conditions, which advanced our understanding of the wave dissipation process in vegetation (WDV) in field conditions. In intertidal environments, waves commonly propagate into vegetation fields with underlying tidal currents, which may alter the WDV process. A number of experiments addressed WDV with following currents, but relatively few experiments have been conducted to assess WDV with opposing currents. Additionally, while the vegetation drag coefficient is a key factor influencing WDV, it is rarely reported for combined wave-current flows. Relevant WDV and drag coefficient data are not openly available for theory or model development. This paper reports a unique dataset of two flume experiments. Both experiments use stiff rods to mimic mangrove canopies. The first experiment assessed WDV and drag coefficients with and without following currents, whereas the second experiment included complementary tests with opposing currents. These two experiments included 668 tests covering various settings of water depth, wave height, wave period, current velocity and vegetation density. A variety of data, including wave height, drag coefficient, in-canopy velocity and acting force on mimic vegetation stem, are recorded. This dataset is expected to assist future theoretical advancement on WDV, which may ultimately lead to a more accurate prediction of wave dissipation capacity of natural coastal wetlands. The dataset is available from figshare with clear instructions for reuse (https://doi.org/10.6084/m9.figshare.13026530.v2; Hu et al., 2020). The current dataset will expand with additional WDV data from ongoing and planned observation in natural mangrove wetlands.

## 1 Introduction

Coastal wetlands, such as mangroves, saltmarshes and seagrasses, are increasingly recognized as effective buffers against wind waves. They can efficiently reduce incident wave height, even in storm conditions (Möller et al., 2014; van Loon-Steensma et al., 2014, 2016; Vuik et al., 2016). Therefore, ecosystem-based coastal defense systems have been proposed as a cost-effective and ecologically sound alternative to conventional coastal engineering (Temmerman et al., 2013; Arkema et al., 2017; Leonardi et al., 2018). These new coastal defense systems have been brought into practice in the Netherlands and the US as 'living shorelines' (Borsje et al., 2017; Currin, 2019), which may be adapted in many other areas around the globe.

Since the first theoretical work by Dalrymple et al. (1984), wave dissipation by vegetation (WDV) has been extensively studied through field surveys (e.g., Jadhav et al., 2013; Vuik et al., 2016; Garzon et al., 2019), laboratory experiments (e.g., Lara et al., 2016; Yao et al., 2018; He et al., 2019; Tinoco et al., 2020), theoretical and numerical models (e.g., Méndez and Losada, 2004; Losada et al., 2016; Hu et al., 2019; Suzuki et al., 2019). Among others, flume and wave basin experiments examining WDV in controlled and repeatable conditions have revealed that WDV is affected both by vegetation canopy traits and hydrodynamic conditions, e.g. water depth, wave period and wave height. The obtained datasets show that increases with vegetation density, stem stiffness and incident wave height (Augustin et al., 2009; Anderson and Smith, 2014), while it decreases with submergence ratio (the ratio between water depth $h$ and canopy height $h_v$, Stratigaki et al., 2011; Maza et al., 2015). Recent experiments introduced more realistic vegetation morphology (He et al., 2019; Maza et al., 2019) and even real vegetation (Ozeren et al., 2014; Lara et al., 2016) to fully reveal the WDV process in natural coastal wetlands.

In intertidal environments, tidal currents generally flow into the vegetation wetlands in the same direction as incident waves during flooding tide and revise during ebb tide. Using wave as a reference, the underlying currents that flow in the same direction as waves are defined as following currents, whereas the underlying currents that flow in the oppose direction as waves are defined as opposing currents. A number of experiments have tested the impact of co-existing following currents on WDV (Li and Yan, 2007; Paul et al., 2012; Hu et al., 2014). They have shown that following currents can both promote and suppress WDV depending on the ratio between imposed current velocity and amplitude of horizontal orbital velocity ($\alpha=U_c/U_w$). As contrast, there are fewer experiments that include opposing currents (Ota et al., 2005; Maza et al., 2015). Maza et al. (2015) conducted a unique experiment in a wave basin to investigate the effect of both following and opposing currents on the WDV of submerged canopies. However, emergent conditions were not included in Maza et al. (2015), which is very like to occur in e.g., tall mangrove forests. Additionally, although recent experiments have improved our understanding of WDV in combined wave-current flows (Losada et al., 2016; Lei & Nepf, 2019), to our knowledge, these experimental datasets are not openly accessible to the research community to foster further advances.

To understand and assess WDV, the knowledge of vegetation drag coefficient ($C_D$) and its variation in different flow conditions is critical. $C_D$ is an empirical parameter that links known velocity ($u$, either from measurements or modeling) to the drag force exerted by vegetation stems ($F_d \sim C_D * u^2$, Morison et al., 1950), which is directly related to WDV. Thus, the determination of $C_D$ is important to accurate WDV assessment. Its variation with characteristic hydrodynamic parameters, i.e., Reynolds number ($Re$) and Keulegan-Carpenter number ($KC$), has been extensively investigated (Nepf, 2011). $C_D$ is commonly derived by calibration method, i.e., calibrating the $C_D$ value to ensure the modeled WDV fits with the observation (e.g., Méndez and Losada, 2004; Li and Yan, 2007; Koftis et al., 2013). A more recent direct measurement method has been proposed to derive $C_D$ via analyzing synchronized $F_d$ and $u$ on the vegetation stems (Hu et al., 2014; Chen et al., 2018). Such a method does not rely on WDV models but is based on the original Morison equation (Morison et al., 1950). Thus, it can avoid potential errors introduced by WDV models and be readily applied in combined current-wave conditions. However, $C_D$ and $F_d$ in combined current-wave flow conditions have been much less reported, especially when waves co-exist with opposing currents. To our knowledge, there is no such dataset available that enables further analysis.

This paper presents a combined dataset composed of two flume experiments on WDV with underlying currents in both emergent and submerged conditions (Hu et al., 2020). These two experiments were conducted in 2014 and 2019, respectively (hereafter referred to as E14 and E19). Both experiments applied stiff wooden cylinders to mimic wooden mangrove canopies. In total, E14 conducted 314 tests, and E19 conducted 354 cases with different scenarios of incident waves, imposed current, vegetation density, and submergence ratio (Table B1). E14 has systematically compared the variations of WDV and $C_D$ with or without co-existing following currents (Hu et al., 2014). As complementary to the E14, E19 further conducted tests with opposing currents. To our knowledge, it is the first freely assessable dataset that includes a wide range of current-wave combinations. Besides wave height variations, this new dataset contains detailed time series data of $F_D$ and $u$ in all the tests and velocity profiles in a few selected tests. These data are essential in assessing $C_D$ and WDV. It is expected to serve future laboratory, theoretical and numerical studies on WDV, which may eventually lead to a more accurate prediction of wave dissipation efficiency of natural coastal wetlands. The potential usage of this dataset and future avenues to advance our understanding are discussed.

## 2 Methods

### 2.1 Flume setup of E14

E14 was conducted in the Fluid Mechanics Laboratory at the Delft University of Technology in 2014 (Hu et al., 2014). The used wave flume was 40 m long and 0.8 m wide (Figure 1a). Currents were imposed in the same direction of the wave propagation, i.e., following currents. We used stiff wooden rods that were fixed vertically on a false bottom as vegetation mimics. The length of the mimic mangrove canopy was 6 m, which was made of wooden rods. The height ($h_v$) and diameter

($b_v$) of the rods was 0.36 m and 0.01 m, respectively. Tested water depth ($h$= 0.25 m and 0.5 m) is chosen to mimic emergent and submerged conditions (Table B1). To avoid complex forcing on vegetation stems, in emergent conditions, the wave crests were always lower than the top of the canopy, whereas in submerged conditions, the wave troughs were always higher than the top of the canopy. In the emergent and submerged conditions, the submergence ratios ($h/h_v$) were 1 and 1.39, respectively. The tested stem densities were $N_v$=62, 139, and 556 stems/m$^2$, denoted as VD1, VD2 and VD3, respectively (Table B1). The mimics were placed following a regular stagger pattern (Figure B1). To measure the wave height attenuation caused by the friction of flume bed and sidewalls, control tests with no mimic stems (VD0) were also tested.

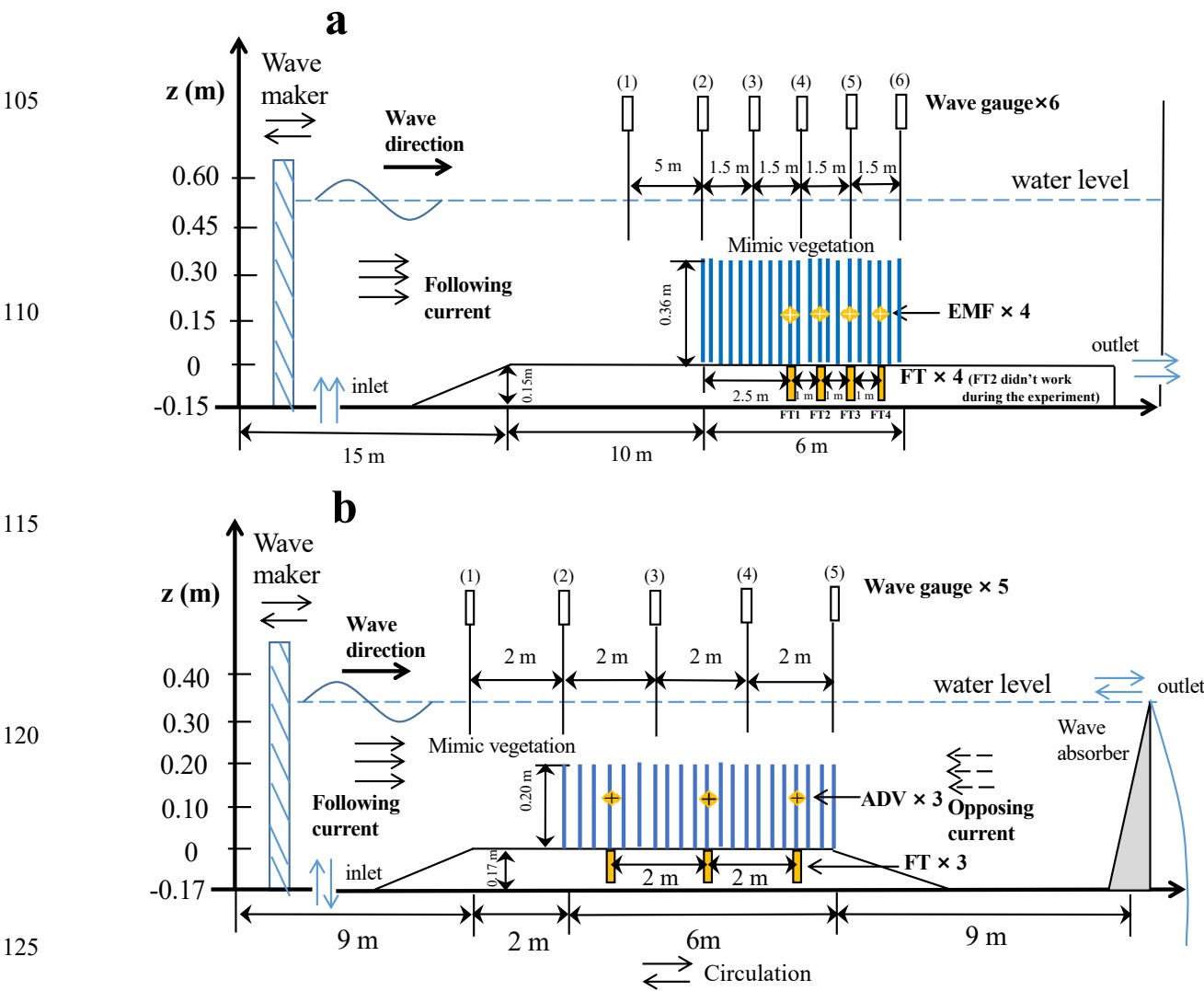

Figure 1. Diagrams of the flume experiments. (a) flume setup of E14, in which waves were imposed either without current or with following currents. EMF is electromagnetic flow manufacture meters for velocity measurements. FT is force transducer that can measure the total force on a mimic stem. (b) flume setup of E19, in which additional tests of waves with opposing currents were included.

In E14, wave height variation was measured by six capacitance-type wave gauges (WG1–WG6) installed in the flume (Figure 1a). The capacitance-type wave gauges were made by Deltares, and its accuracy was ±0.5% (Delft Hydraulics, 1990). Force transducers (FT1-4) were installed to measure the acting force $F$ on four individual vegetation mimics along with the canopy (Figure 1a and Figure A1). To minimize disturbance to the flow, all the FTs were installed underneath the false bottom. FT1 and FT3 were developed by Deltares, the Netherlands, whereas FT2 and FT4 were force transducers made by UTILCELL (model 300). The output of FTs is in voltage, and it can be converted to acting force in both positive and negative directions by linear regressions. The calibration was done similarly to Stewart (2004). The output value does not change with the positions of the forcing on the attached vegetation mimics, i.e., the same force gives the same value no matter where the force is acting on the mimics. Force data were sampled at 1000 Hz to capture force variation within a wave period. The accuracy of the FTs was estimated to be ±1%, and more details on the FTs can be found in Bouma et al. (2005). FT2 (the 2nd one in the wave direction) failed during the experiment, data from which were excluded for analysis.

Velocity ($u$) was measured at half water depth by EMFs (electromagnetic flow manufacture meters) made by Deltares (accuracy ±1%, Delft Hydraulics, 1990). Four EMFs were installed at the same cross-sections as the force transducers to obtain in-phase horizontal velocity (Figure 1a), and subsequently used to derive vegetation drag coefficient ($C_D$). The deriving method is detailed in Appendix C. The velocity measurement was to obtain representative in-canopy velocities. Thus, in submerged canopies, it was perhaps more suitable to measure velocity at half of the canopy height than at half water depth. However, given the relatively shallow water depths tested in both E14 and E19, velocities obtained at both positions were similar, as shown in the vertical velocity profiles (see Figure 4). These vertical velocity profiles were measured in a few selected cases (see Appendix B). It was done by moving the measuring probes vertically in repeat experiment runs. The velocity profiles were measured in the vegetation canopies far away from both ends of the flumes, to avoid the potential local influence of the in- and outlets.

## 2.2 Flume setup of E19

E19 was conducted in the Coastal Dynamics Laboratory at Sun Yat-Sen University. As a complement to E14, E19 included cases of pure wave, wave with following currents, and additional cases of wave with opposing currents. It was conducted in a 26 m long, 0.6 m wide, 0.6 m high wave flume (Figure 1b). Currents were imposed in the same and opposite direction as the wave propagation. We adapted the same vegetation canopy width and diameter as the E14. The main differences of the mimic mangrove canopy were: 1) the mimic canopy was 0.25 m tall; 2) low-density case (VD1) of E14 was excluded, whereas VD0, VD2 and VD3 cases of E14 were retained in the E19; 3) additional tests with randomly arranged mimics (VD2R, VD3R) were included (Figure B1); 4) two water depths ($h$=0.2/0.33 m) were chosen to mimic emergent and submerged canopies (submergence ratio $h/hv$ = 1 and 1.32, Table B1).

Three FTs were installed to measure $F$ acting on vegetation mimics (Figure 1b). These FTs were model M140 made by UTILCELL with an accuracy of ±1.3% (https://www.utilcell.com/en/load-cells/load-cell-m140; Hu et al., 2020). These FTs were mounted in the false bottom to avoid disturbance of the flow. Their output was in mass and it can be converted to force by multiplying the acceleration of gravity. The measuring rods on FTs were made of stainless steel, so that they can be fixed tightly to the FTs (Figure A1). $F$ was sampled at 50 Hz. Velocity ($u$) was measured by 3 ADVs (acoustic doppler velocimeter) at the same cross-sections of FTs in the canopy (Figure 1b). They were made by Nortek with an accuracy of ±0.5% (https://www.nortekgroup.com/products/vectrino; Hu et al., 2020). Similar to E14, $u$ was measured at half of the water depth at 50 Hz. In a few selected tests, velocity profiles were obtained by moving the ADV probe vertically (see Appendix B).

## 2.3 Wave conditions in E14 and E19

In both experiments, the tested waves were regular waves. The tested wave height was 0.04-0.2 m, and the wave period was 0.6-2.5 s (see Table B1). We defined the direction of wave propagation as 'positive' direction and the opposing direction as 'negative' direction. Due to Doppler Effect, the wave height could be reduced or increased when waves propagate with following and opposing currents (Demirbilek et al., 1996). For tests with the same wave conditions but different co-existing currents, we adjusted the wave input to ensure the wave height arrived at the vegetation front is similar in each test with different co-existing current velocity (within 5%). This treatment is to 1) avoid possible influence caused by different incident wave height, and 2) reflect field conditions with similar incident wave heights but with various underlying tidal currents (Garzon et al., 2019). In each test, the water depth and discharge were set to the targeted values to create steady currents. Waves were imposed after the steady currents and water levels were achieved. To avoid the complex wave reflection conditions, we only analyzed the first 3-5 waves after the spinning up waves. We turned off the wave-makers after about 20 waves in each test.

It is noted that the imposed waves in both experiments were not strictly linear but contained small nonlinear components. This nonlinearity leads to weak recirculation in the flume, which can be observed from the negative in-canopy velocity in pure wave cases (Figure 4). This recirculation in the flumes is common in wave flumes and attributed to Stokes drift (Hudspeth & Sulisz, 1991). The effect of this nonlinearity and recirculation on WDV has been discussed in Hu et al. (2014). Additionally, this recirculation can also occur in field conditions as wetlands are often bounded by landward dikes. These dikes are closed boundaries similar to the baffle plates in confined flumes, which can also induce Stokes drifts. Lastly, the impact of bottom and sidewall friction can be observed in control tests without vegetation (VD0) and documented in the dataset.

## 2.4 Data analysis

In both experiments, we measured spatial wave height change, time series of acting force on vegetation mimic ($F$) and velocity at the middle water depth ($u$) as an approximation of the depth-averaged velocity (see Figure 4). Following Morison equation (Morison, 1950), $F$ on a vegetation mimic can be specified as:

$$F = F_D + F_M = \frac{1}{2}\rho C_D h_v b_v u|u| + \frac{\pi}{4}\rho C_M h_v b_v^2 \frac{\partial u}{\partial t} \tag{1}$$

$F_D$ and $F_M$ are drag force and inertia force, respectively. $C_M$ is the inertia coefficient, which value is equal to 2 for cylinders (Dean and Dalrymple, 1991). $\rho$ is the density of water. $u$ is the depth-averaged horizontal flow velocity, and it is assumed to be equal to the flow velocity at half water depth (Hu et al., 2014). Using known $u$ and $C_D$, $F$ can be reproduced by Eq. (1). $u$ can be decomposed as:

$$u(t) = U_{mean} + U_w \sin(\omega t) + U' \tag{2}$$

where $\omega$ is the wave angular frequency, $U'$ is turbulent velocity fluctuations, which is neglected in the analysis for simplicity. $U_{mean}$ is the averaged velocity over a wave period ($T$), defined as (e.g. Pujol et al., 2013):

$$U_{mean} = \frac{1}{T}\int_0^T U(t)dt \tag{3}$$

Please note that $U_{mean}$ is not equal to $U_c$, which is the imposed current velocity without the influence of waves. $U_w$ is the amplitude of the horizontal wave orbital velocity and can be defined as:

$$U_w = \frac{1}{2}(u_{max} - u_{min}) \tag{4}$$

where $u_{max}$ and $u_{min}$ are the measured peak flow velocities in the positive and negative directions in a wave period ($T$). Both $u_{max}$ and $u_{min}$ change with co-existing mean currents. To accommodate empirical $KC$-$C_D$ relations, $KC$ number is defined as following (Keulegan and Carpenter, 1958; Chen et al., 2018):

$$KC = \frac{Max(|u_{max}|, |u_{min}|) * T}{b_v} \tag{5}$$

Wave height ($H$) along the mimic vegetation canopy can be descried as:

$$K_v = \frac{H}{H_0} = \frac{1}{1 + \beta x} \tag{6}$$

$H_0$ is the wave height at the canopy front. $x$ is the distance into the canopy and $\beta$ is a damping coefficient, which can be obtained by fitting Eq. (6). To reveal the effect of co-existing currents, the relative wave height decay in current-wave and wave-only case $r_w$ is defined as:

$$r_w = \frac{\Delta H_{cw}}{\Delta H_{pw}} \tag{7}$$

where the $\Delta H_{pw}$ and $\Delta H_{cw}$ are the wave height reduction in pure wave and current-wave cases.

## 3 Data

### 3.1 wave dissipation in vegetation canopy with following and opposing currents

For pure wave cases, WDV in both experiments has similar variation. Emergent and denser canopies result in greater WDV than submerged and sparser canopies (Figure 2a and 1b). Additionally, such variation can also be found in the randomly distributed vegetation canopy. No apparent difference can be found between regular and random canopies (Figure 2c). In waves plus following current cases, the two experiments also show similar results in WDV (Figure 2d and 2e). When the following current is small (0.05 m/s for E14 and 0.03 m/s for E19), the accompany current slightly reduces WDV comparing

to the pure wave cases. However, as the following current velocity increases (0.15 m/s for E14 and 0.12 m/s for E19), WDV is increased compared to the pure wave cases. WDV may be further enhanced by a stronger following current (0.20 m/s for E14 and 0.15 m/s for E19). As a contrast, opposing currents immediately increase WDV even when the velocity magnitude is small (Figure 2f). As the opposing current velocity increases, the WDV is promoted to a higher level comparing to the cases with the following currents.

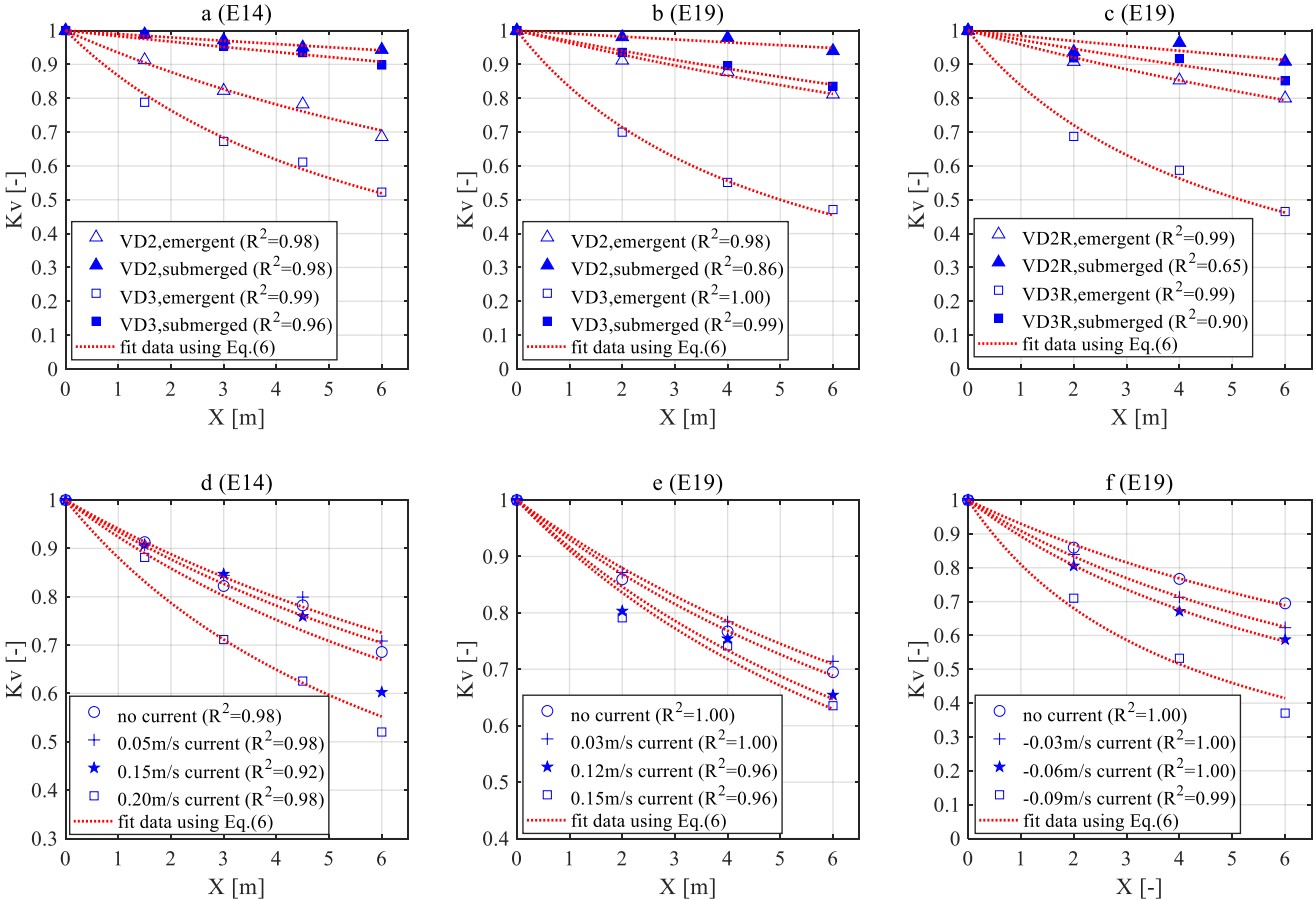

**Figure 2. Relative wave height ($K_v$) variation through vegetation canopies ($X$=0-6 m). (a) $K_v$ reduction by regular vegetation mimics in pure wave conditions in E14. The tested wave height is 4 cm and wave period is 1.0 s (i.e. wave0410); (b) $K_v$ reduction by regular vegetation mimics in pure wave conditions in E19. The tested wave condition is wave0308; (c) $K_v$ reduction by randomly disputed vegetation mimics in pure wave conditions in E19. The tested wave condition is wave0308; (d) $K_v$ reduction with following currents in E14. The tested wave condition is wave0410; (e) $K_v$ reduction with following currents in E19. The tested wave condition is wave0510; (f) $K_v$ reduction with opposing currents in E19. The tested wave condition is wave0510. Note the different scale of the Y-axis in d-f.**

The results of the two experiments present a synthesis of WDV variation with underlying currents (Figure 3). In cases with the following currents, the relative wave height decay ($r_w$, ratio of wave height decay between current-wave and wave-only case) has a similar variation in E14 and E19. When $\alpha$ is in the range of [0 1], $r_w$ is generally lower than 1, i.e., WDV is suppressed compared to the pure wave cases. As contrast, when $\alpha$ is larger than 1, $r_w$ is generally larger than 1, i.e., WDV is enhanced

instead. Notably, negative $\alpha$ leads to higher $r_w$ compared to positive $\alpha$ with the same magnitude. Thus, opposing currents can more easily increase WDV compared to the following currents. Notably, $r_w$ value can reach 4-5 with both following and opposing currents, highlighting the impact of underlying currents on WDV.

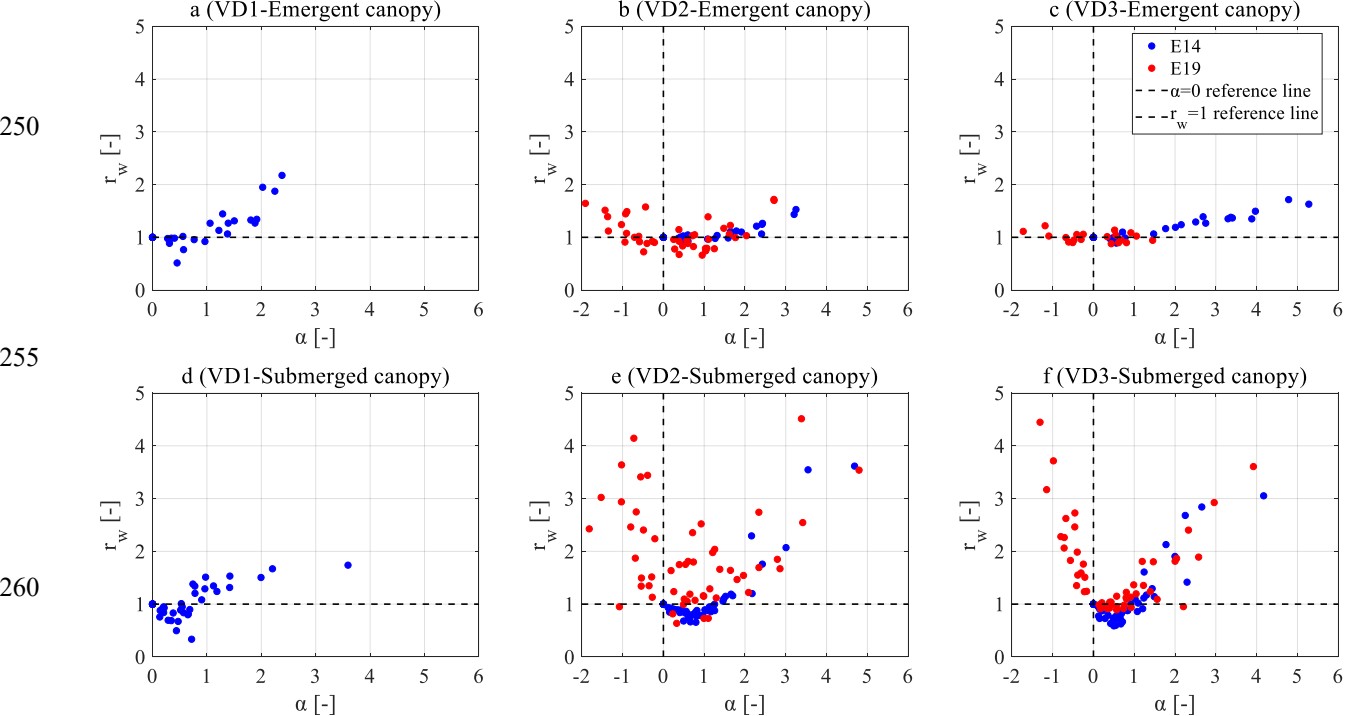

Figure 3. Relation between velocity ratios $\alpha$ and the relative decay $r_w$. (a), (b) and (c) show the variation of $r_w$ with $\alpha$ in emergent canopies with stem densities of VD1, VD2 and VD3, respectively. (d), (e) and (f) show the variation of $r_w$ with $\alpha$ in submerged canopies with stem densities of VD1, VD2 and VD3, respectively. The E14 data points are redrawn from Hu et al., (2014) with permission of Elsevier.

## 3.2 Velocity and force data

Since the variation of WDV in different flow conditions is closely related to the spatial velocity structures, we measured the vertical velocity profiles in a few tests with the same wave condition but different accompany currents (Figure 4). Velocity profiles reveal a significant difference in flow structures between cases with various submergence and co-existing current conditions. A few similar patterns can be observed from both experiments: 1) the direction of $U_{mean}$ is determined by the imposed current velocity; 2) in submerged canopies with co-existing currents, a distinctive velocity shear layer can be observed near the top of the vegetation canopy, whereas in emergent canopies velocity profiles are generally uniform; 3) the existence of vegetation reduces $U_{mean}$ magnitude comparing to the control VD0 case. 4) when comparing wave-only and wave-current cases, the presence of wave leads to lower $U_{mean}$ magnitude, regardless of the direction of the currents; 5) negative $U_{mean}$ can be found in pure wave condition, which plays an important role in WDV variation as pointed out in the theoretical model in

Hu et al., (2014). The presented velocity profiles are similar to previous experiments (e.g., Li and Yan, 2007; Pujol et al., 2013).

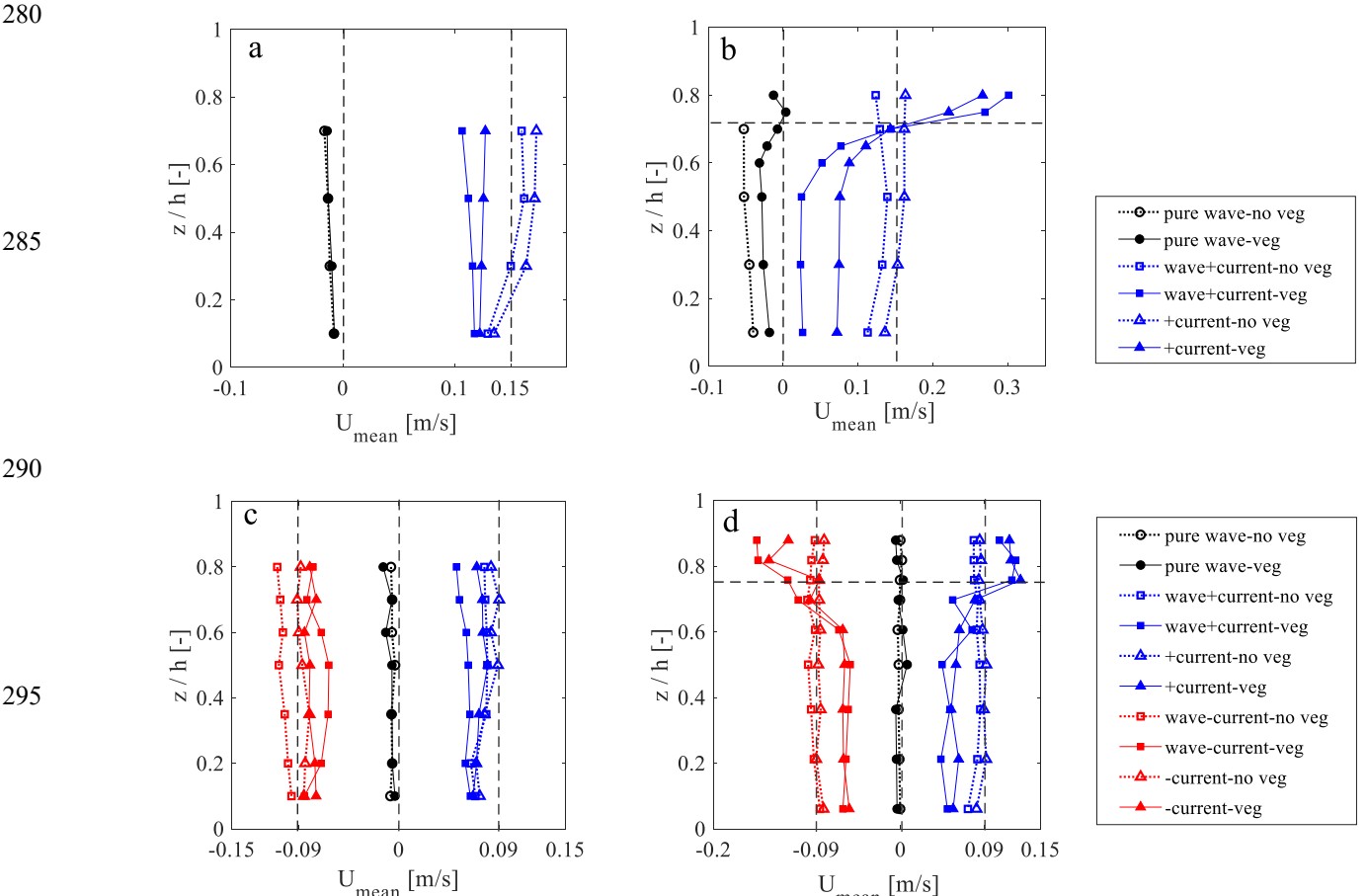

Figure 4. Vertical profile of time-mean velocity ($U_{mean}$). (a) emergent canopy with incident wave height of 6 cm and wave period of 1.2 s (i.e. wave0612) in E14. The vertical dash lines indicate the imposed current velocities; (b) submerged canopy with case wave1518 in E14. The horizontal line indicates the top of the vegetation canopy; (c) emergent canopy with case wave0508 in E19; (d) submerged canopy with case wave0508 in E19. The E14 data points are redrawn from Hu et al., (2014) with permission from Elsevier.

Apart from the vertical velocity structures, we also include the raw data of the temporal variations of velocity ($u$) and the acting force ($F$) on vegetation mimics at multiple locations along vegetation canopies to derive $C_D$ for all the tested cases (Figure 5). In each test, velocity and force measurements were taken at the same cross-sections. However, time lags still exist between the velocity and force data, which can be perceived via the phase difference between $u$ peak and drag force peak (Figure 5d). These time lags may be induced by small misalignments between the ADV probes and the force transducers, as well as the intrinsic delays of these instruments. To reduce the time lags and facilitate deriving $C_D$, an automatic algorithm is applied to synchronize $u$ and $F$ data, i.e., reducing the time lags between the peaks of $u$ and $F_D$ (Figure 5e). As a validation of the synchronization, the computed $F_D$ (using derived $C_D$) and $F_M$ signals are used to compose a reproduced $F$, which is

subsequently compared with the measured total force. A comprehensive comparison shows that the calculated *F* is consistent with the measured total force (see Figure C1).

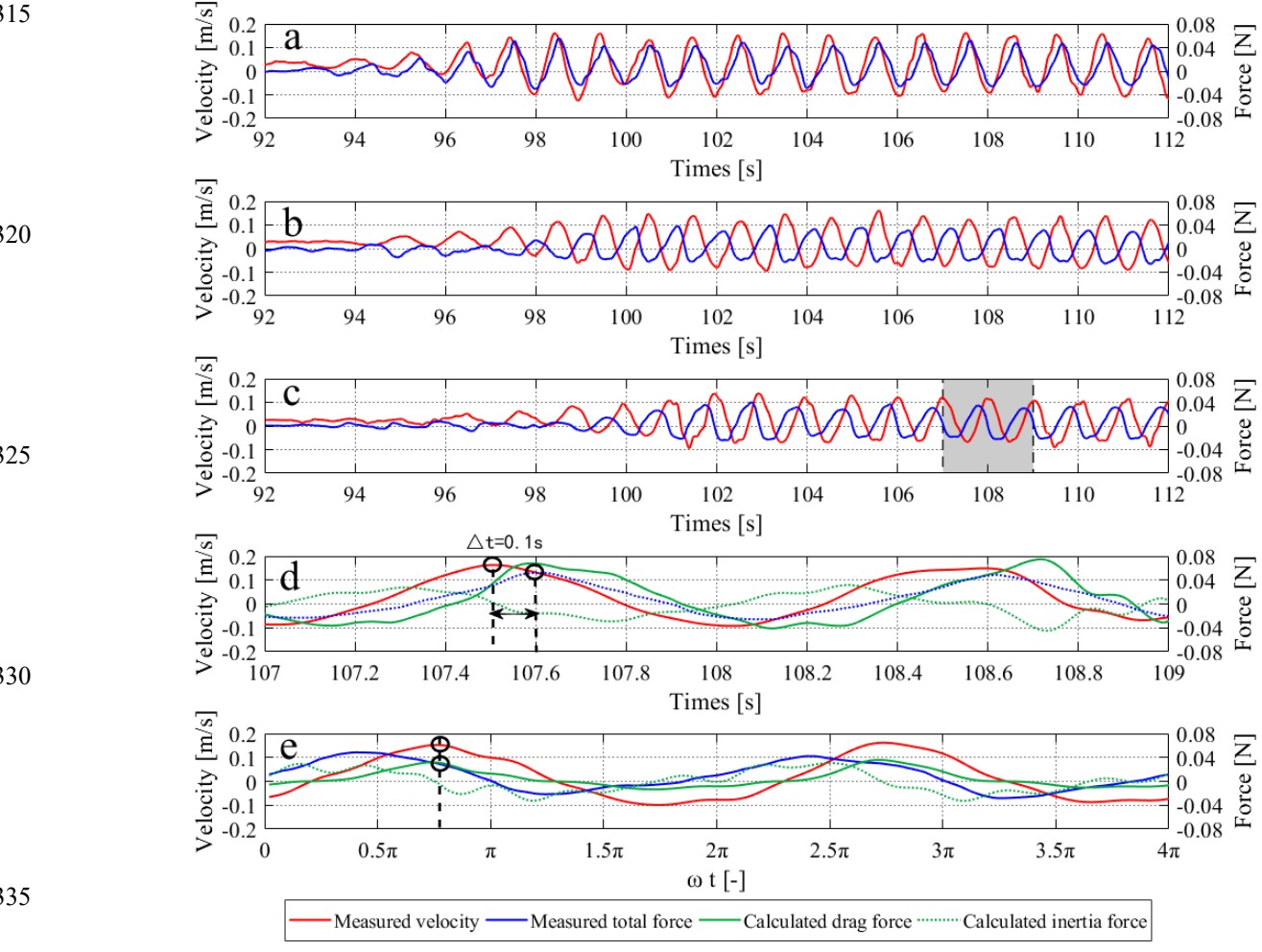

**Figure 5. Synchronized velocity and force time series. (a-c) measured raw velocity and total force data at three locations in E19 in the direction of wave propagation; (d) enlarged data of the shaded area of (c), which shows the time shift (*Δt*) between *u* and *F_D* is about 0.1 s. (e) synchronized *u* and *F_D* data, which are processed following the method of Yao et al., (2018). The shown test case is with 5 cm wave height, 1.0 s wave period and 0.03 m/s following current.**

### 3.3 Drag coefficients

Our combined dataset shows an overall reduction trend of $C_D$ with *KC* number across all the conditions of vegetation density, submergence ratio, and co-existing currents (Figure 6). In E19, $C_D$ reduces fast when *KC* increases from close to zero to 10. When the *KC* number approaches 20, $C_D$ is reduced quickly to about 2. As the KC number rises above 20, $C_D$ further reduces and finally reaches a nearly constant value of 1.30. It is noted that the variation of $C_D$ in opposing currents is similar to that of

the following currents. There is no apparent difference between the two experiments, except that E14 contains a wider *KC* range than E19 (Figure 6b). A *C_D-KC* relation for combined E14 and E19 data is listed below:

$$C_D = 0.95 + 11.39KC^{-1.09}, R^2 = 0.72 \tag{8}$$

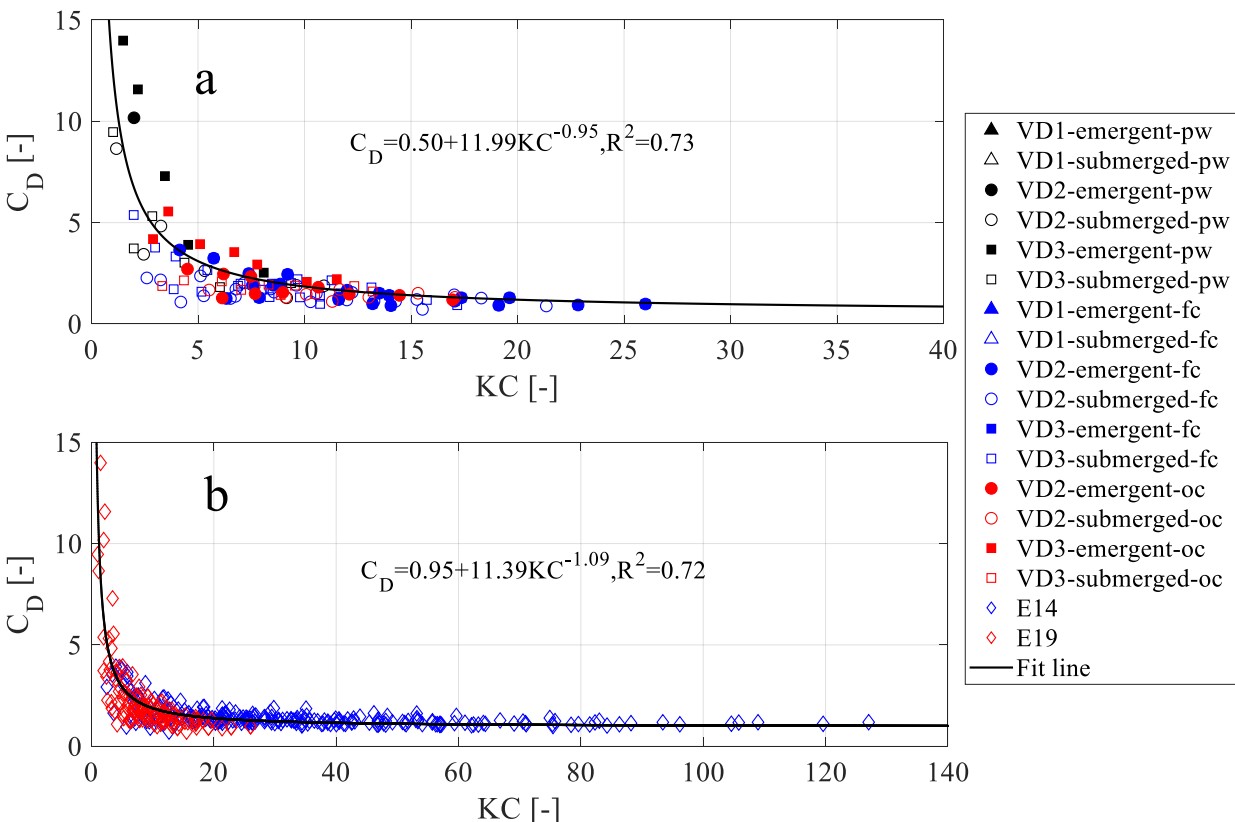

Figure 6. Relation between *KC* and *C_D*. (a) *C_D* in E19 with cases of pure wave ('pw'), wave with following current ('fc') and wave with opposing current ('oc'); (b) combined *C_D* in both E14 and E19. *C_D* were derived using the direct measurement approach (Appendix C).

## 4 Recommendations for Data Reuse

### 4.1 Towards a uniform drag coefficient relation

Our dataset includes a wide range of *C_D* in pure wave and wave-current flows. Base on such dataset, we derived a uniform *C_D-KC* empirical relation covering various combined wave-current conditions with both following and opposing currents. We reveal that *C_D* in opposing currents is also negatively correlated to *KC*, similar to other flow conditions. The *C_D* data with opposing currents are new supplementary to the existing studies. The resulting empirical relation can be valuable to the modelling of WDV studies, especially those considering underlying currents. (Henry et al., 2015; Hu et al., 2019; Suzuki et

al., 2019; van Veelen et al., 2021). When velocities are unknown to define $KC$ numbers, the velocities may be estimated by linear wave theory or by numerical iterations. For the latter case, an initial $C_D$ value can be set as 1 to start the iteration. The current dataset also includes in-canopy velocity, acting force and temporally varying $C_D$. These data can be useful in assessing the force on vegetation stems and estimating e.g. survival of a mangrove canopy in storm events. Lastly, as our experiments have tested numerous cases with varying canopy density, water depth and current-wave conditions, the generated dataset is thus suitable for machine learning quest, as such an approach can be capable of deriving more sophisticated relations from multidimensional and nonlinear data (Tinoco et al., 2015; Goldstein et al., 2019).

## 4.2 A unique dataset for further researches in WDV

Our experiments provide a unique dataset of wave height variation through vegetation with co-existing following and opposing currents. It shows that co-existing currents have a substantial impact on WDV. They can reduce WDV by nearly 50% or increase WDV by four times depending on the current velocity ratio ($\alpha$). Thus, the effect of currents should account for inaccurate WDV assessment. Our data reveal two general patterns of the wave dissipation trend with co-existing currents. First, WDV is suppressed or not sufficiently enhanced when the co-existing current velocity is small, but it is promoted when the current velocity is high, regardless of the imposed velocity direction. Second, in submerged canopies, opposing currents are more likely to promote WDV compared to the following currents. Notably, cases with weak following currents have the lowest WDV in both experiments. Therefore, to ensure safety, these cases should be regarded as the critical condition in designing nature-based coastal defense projects.

For simplicity, the presented dataset does not include tests of flexible vegetation (e.g., saltmarshes and seagrass, e.g., Luhar and Nepf, 2011; Maza et al., 2015; van Veelen et al., 2020; 2021) nor vegetation with root or leaves (He et al., 2019; Maza et al., 2019). We expect that the present dataset will expand with additional WDV data in natural mangrove wetlands from ongoing and future observation. While future experiments can certainly benefit from more realistic vegetation characteristics, the current dataset is still valuable in supporting the development of theoretical and numerical models (Losada et al., 2016; Suzuki et al., 2019), as the simplified setting of vegetation canopy facilitates in-depth investigation of complex wave-current-stem interactions. In fact, the $C_D$ relation derived in E14 has already been successfully applied in modeling wave dissipation by real flexible marsh plants, i.e., *S. Anglica*, *P. Maritima* and *E. Athericus* (van Veelen et al., 2021). This indicates that the application range of the present dataset is not limited to rigid artificial vegetation but can also be extended to flexible real vegetation. Thus, the present dataset may aid the assessment of the wave dampening capacity, coastal vegetation wetlands as a measure for coastal defense.

## 5 Data availability and future observations

All data presented in this paper are available from figshare (https://doi.org/10.6084/m9.figshare.13026530.v2; Hu et al., 2020). The repository includes data as well as instructions in readme files. Additionally, we expect that the current repository will

expand with additional WDV data from ongoing and planned future observation in real mangrove wetlands, e.g. from ANCODE project (https://www.noc.ac.uk/projects/ancode).

**Acknowledgments and Data**

This work is supported by ANCODE (Applying nature-based coastal defense to the world's largest urban area—from science

to practice) project, a three-way international funding through the Chinese National Natural Science Foundation (NSFC, Grant 51761135022), the Netherlands Organization for Scientific Research (NWO, lead funder, Grant ALWSD.2016.026), and the U.K. Research Councils (UKRI Grant EP/R024537/1), a project from the National Natural Science Foundation of China (No. 51609269) and Guangdong Provincial Department of Science and Technology (2019ZT08G090). The data presented in this paper is freely accessible at https://doi.org/10.6084/m9.figshare.13026530.v2.

**Author contribution**

ZH, LS, HW and YL conducted the experiments and collected the raw data. ZH, MS and TS designed the experiments. ZH, LS and YL prepared the manuscript with contributions from all authors.

**Competing interests**

The authors declare that they have no conflicts of interest.

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

## Appendix A. Photos of the experiment instruments and setup

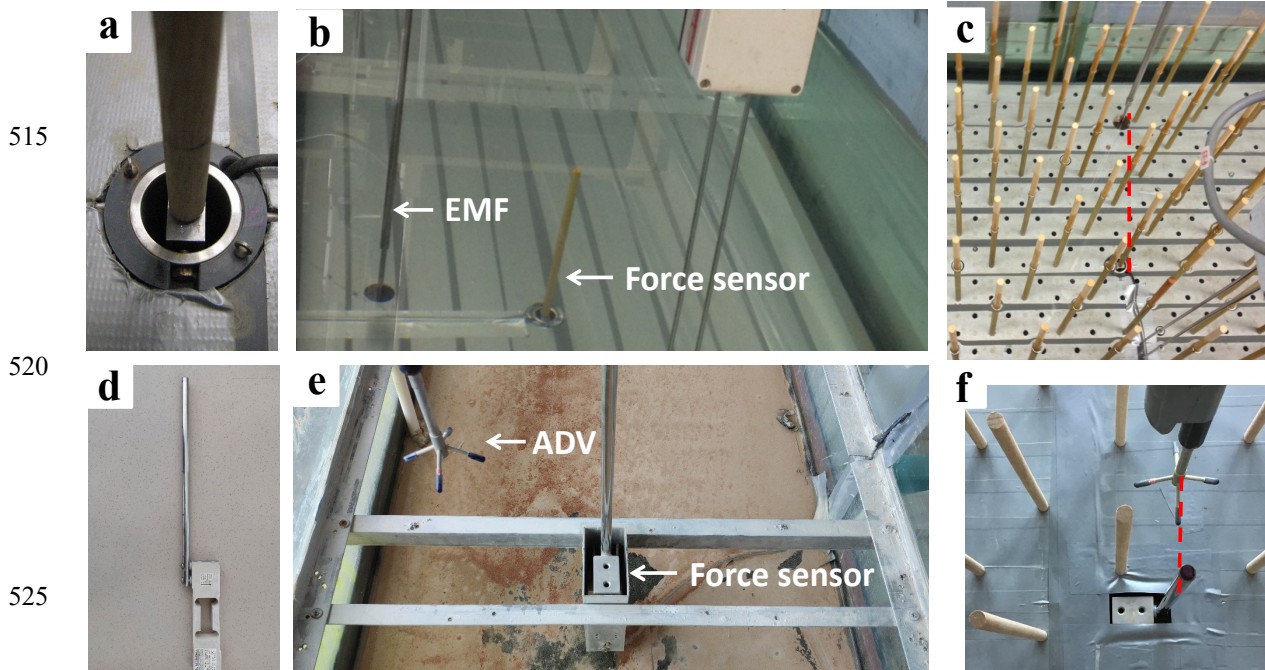

**Figure A1. Photos of the applied instruments and canopy arrangement in E14 (a-c) and E19 (d-f). In E14, (a) force transducer and (b) EMFs (electromagnetic flow manufacture meters) for velocity measurement were developed by Deltares (former Delft Hydraulics, the Netherlands). (d) force transducer (model M104) developed by UTILCELL and (e) ADVs (acoustic doppler velocimeter) for velocity measurement were from Nortek. (c) and (f) show that the force and velocity measurements were taken at the same transect of the flume to obtain synchronized data.**

## Appendix B. Test conditions in the two experiments

Table B1 shows the tested cases in both E14 and E19. A large number of tests were included in both experiments: 314 in E14 and 366 in E19. In all the tests, the wave height spatial variation, in-canopy force and velocity were measured. Each test was conducted at least twice to ensure reproducibility. For a few selected cases, the velocity profiles were measured by moving the EMF or ADV measuring probe vertically in the water column.

In E14, the selected cases were wave0612 and wave1518. For emergent canopy cases (h=0.25 m), the velocity was measured at 4 locations: z/h=0.1, 0.3, 0.5 and 0.7. In submerged canopy cases (h=0.50m), u was measured at 8 locations: z/h=0.1, 0.3, 0.5, 0.6, 0.65, 0.75, 0.8 and 0.9. The measuring location was refined near the top of the canopy ($h_v/h = 0.72$). In E19, the selected cases were wave0508. For emergent canopy cases (h=0.20 m), the velocity was measured at 7 locations: z/h=0.2, 0.3, 0.4, 0.5, 0.65, 0.75 and 0.9. In submerged canopy cases (h=0.33m), u was measured at 9 locations: z/h=0.12, 0.18, 0.24, 0.30, 0.39, 0.5, 0.63, 0.79 and 0.94.

**Table B1. Test conditions in E14 and E19 with different combinations of hydrodynamic conditions and mimic canopy configurations**

| Source | Water depth ($h$)/plant height ($h_v$) | Stem density (N) [#/m²] | Wave height ($H$) [m] | Wave period ($T$) [s] | Wave case | Co-existing current velocity direction and magnitude ($U_c$) [m/s] |
|---|---|---|---|---|---|---|
| | | 62/139/556 | 0.04 | 1.0 | Wave0410[a] | 0/+0.05/+0.15/+0.20 |
| | | 62/139/556 | 0.04 | 1.2 | Wave0412 | 0/+0.05/+0.15/+0.20 |
| | | 62/139/556 | 0.06 | 1.0 | Wave0610 | 0/+0.05/+0.15/+0.20 |
| | 0.25/0.36 | 62/139/556 | 0.06 | 1.2 | Wave0612 | 0[c]/+0.05/+0.15[c]/+0.20 |
| | | 62/139/556 | 0.08 | 1.2 | Wave0812 | 0/+0.05/+0.15/+0.20 |
| | | 62/139/556 | 0.08 | 1.5 | Wave0815 | 0/+0.05/+0.15/+0.20 |
| | | 62/139/556 | 0.10 | 1.5 | Wave1015 | 0/+0.05/+0.15/+0.20 |
| E14 | | 62/139/556 | 0.04 | 1.0 | Wave0410 | 0/+0.05/+0.15/+0.20/+0.30[b] |
| | | 62/139/556 | 0.06 | 1.2 | Wave0612 | 0/+0.05/+0.15/+0.20/+0.30 |
| | | 62/139/556 | 0.08 | 1.4 | Wave0814 | 0/+0.05/+0.15/+0.20/+0.30 |
| | | 62/139/556 | 0.10 | 1.6 | Wave1016 | 0[c]/+0.05/+0.15[c]/+0.20/+0.30 |
| | | 62/139/556 | 0.12 | 1.6 | Wave1216 | 0/+0.05/+0.15/+0.20/+0.30 |
| | 0.50/0.36 | 62/139/556 | 0.12 | 1.8 | Wave1218 | 0/+0.05/+0.15/+0.20/+0.30 |
| | | 62/139/556 | 0.15 | 1.6 | Wave1516 | 0/+0.05/+0.15/+0.20/+0.30 |
| | | 62/139/556 | 0.15 | 1.8 | Wave1518 | 0[c]/+0.05/+0.15[c]/+0.20/+0.30 |
| | | 62/139/556 | 0.15 | 2.0 | Wave1520 | 0/+0.05/+0.15/+0.20/+0.30 |
| | | 62/139/556 | 0.18 | 2.2 | Wave1822 | 0/+0.05/+0.15/+0.20/+0.30 |
| | | 62/139/556 | 0.20 | 2.5 | Wave2025 | 0/+0.05/+0.15/+0.20/+0.30 |
| | | 139/556 | 0.03 | 0.6 | Wave0306 | 0/±0.03/±0.06/±0.09/±0.12/±0.15 |
| | | 139/556 | 0.03 | 0.8 | Wave0308 | 0/±0.03/±0.06/±0.09/±0.12/±0.15 |
| | 0.20/0.25 | 139/556 | 0.05 | 0.6 | Wave0506 | 0/±0.03/±0.06/±0.09/±0.12/±0.15 |
| | | 139/556 | 0.05 | 0.8 | Wave0508 | 0[c]/±0.03/±0.06/±0.09[c]/±0.12/±0.15 |
| | | 139/556 | 0.05 | 1.0 | Wave0510 | 0/±0.03/±0.06/±0.09/±0.12/±0.15 |
| E19 | | 139/556 | 0.03 | 0.6 | Wave0306 | 0/±0.03/±0.06/±0.09/+0.12/+0.15 |
| | | 139/556 | 0.03 | 0.8 | Wave0308 | 0/±0.03/±0.06/±0.09/+0.12/+0.15 |
| | | 139/556 | 0.05 | 0.6 | Wave0506 | 0/±0.03/±0.06/±0.09/+0.12/+0.15 |
| | 0.33/0.25 | 139/556 | 0.05 | 0.8 | Wave0508 | 0[c]/±0.03/±0.06/±0.09[c]/+0.12/+0.15 |
| | | 139/556 | 0.05 | 1.0 | Wave0510 | 0/±0.03/±0.06/±0.09/+0.12/+0.15 |
| | | 139/556 | 0.07 | 0.8 | Wave0708 | 0/±0.03/±0.06/±0.09/+0.12/+0.15 |
| | | 139/556 | 0.07 | 1.0 | Wave0710 | 0/±0.03/±0.06/±0.09/+0.12/+0.15 |

[a] wave0410 means the incident wave height is 4 cm and the wave period is 1.0 s.

[b] '+' means current flow in the same direction of waves, '-' means current flow in the opposite direction of waves; in E14, the low vegetation density tests (62 stems/m²) does not have '+0.30 m/s' cases.

[c] in these cases, we conducted velocity profile measurements.

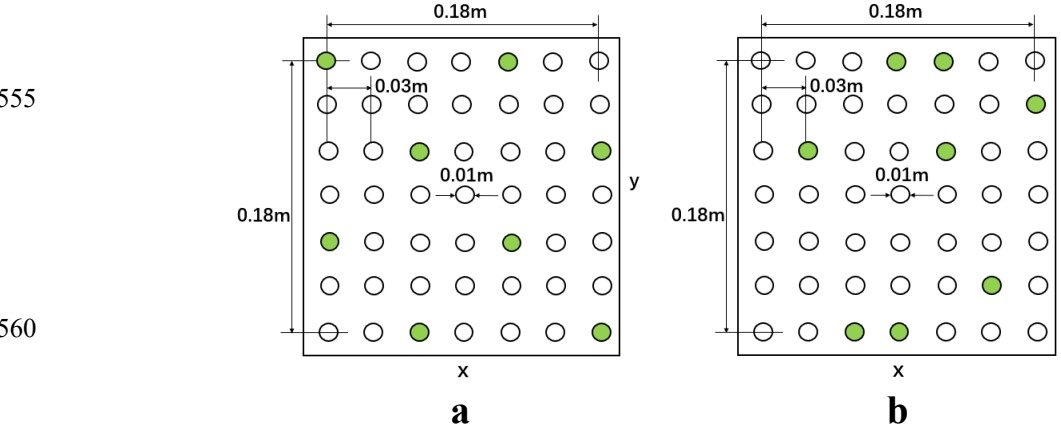

**Figure B1.** top view of vegetation mimics distribution in E19 (a) regular canopy, 139 stems/m²; (b) random canopy, 139 stems/m²

### Appendix C. Direct measurement method of $C_D$

The direct measurement method of $C_D$ in combined current-wave flows was first introduced in Hu et al., (2014) and it was further improved in Yao et al., (2018). Such method is proposed for both pure wave and combined wave-current flows. The force acting on an individual mimic stem is composed of drag force and inertia force, as expressed by Morison equation (Eq. 1, Morison et al., 1950)

The only unknown parameter in Morison equation is drag coefficient $C_D$. To derive period-averaged $C_D$, the direct measurement method applies the technique of quantifying the work done by the acting force (Hu et al., 2014). The work done by the acting force on mimic stem over a full wave period is composed of the work done by the drag force and the inertia force, expressed as:

$$W = W_D + W_M = \frac{1}{T}\int_0^T F_D u\, dt + \frac{1}{T}\int_0^T F_M u\, dt \tag{C1}$$

where $W_D$ and $W_M$ are the work performed by $F_D$ and $F_M$ over a wave period, respectively. Since $W_M$ equals to zero in both pure wave and current-wave conditions, $F_M$ doesn't contribute to the WDV (Dalrymple et al., 1984). Hence $W$ equals to $W_D$. Therefore, the period-averaged $C_D$ can be derived based on the following equation:

$$C_D = \frac{2\int_0^T F_D u\, dt}{\int_0^T \rho h_v b_v u^2 |u|\, dt} = \frac{W_D}{\int_0^T \rho h_v b_v u^2 |u|\, dt} = \frac{W}{\int_0^T \rho h_v b_v u^2 |u|\, dt} = \frac{2\int_0^T F u\, dt}{\int_0^T \rho h_v b_v u^2 |u|\, dt} \tag{C2}$$

Before applying direct measurement to derive $C_D$, the force data and velocity data should be aligned (Figure 5d). Detailed procedure of alignment can be found in Yao et al., (2018). As drag force ($F_D$) is a function of velocity ($u$) Eq. (1), $F_D$ and $u$ should be in the same phase. By using measured total force ($F$), measured velocity ($u$) and the inertia coefficient ($C_M$) into Eq. (1), we can obtain the drag force ($F_D$) and then adjust the phase shift ($\Delta t$) between the velocity and drag force peaks. The obtained new velocity and force data time series will be used as inputs in the next run. This loop is excecuted over 30 times.

Finally, the minimum phase shift ($\Delta t$) and the aligned velocity and force timeseries will be chosen as outputs for deriving $C_D$. As a validation of the directly derived $C_D$, we reproduced the maximum force ($F_{cal\text{-}max}$) in both positive and negative directions using the derived $C_D$, and compared it with the measured maximum force ($F_{mea\text{-}max}$, see Figure C1).

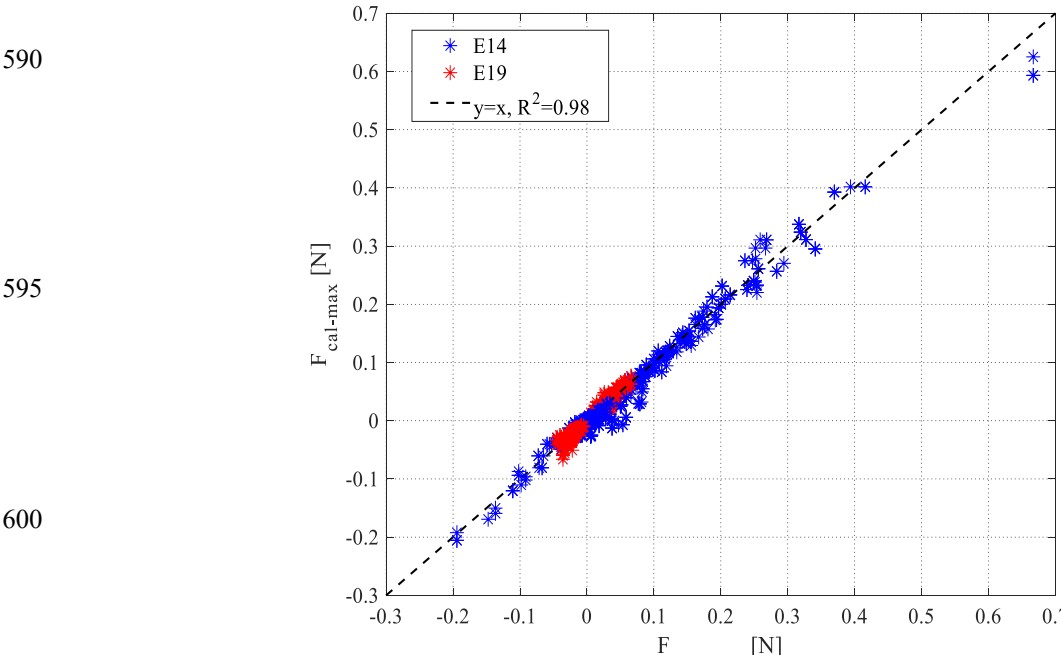

**Figure C1. A comparison between measured maximum force ($F_{mea\text{-}max}$) and calculated maximum force ($F_{cal\text{-}max}$) in both**
**positive and negative directions. $F_{cal\text{-}max}$ is reproduced using directly derived $C_D$.**