# Peer review of "Laboratory data on wave propagation through vegetation with following and opposing currents"

_Earth System Science Data, 2021_

## Author Comment (AC1)

**Reply to RC1 regarding manuscript:**

**Laboratory data on wave propagation through vegetation with following and opposing currents (essd-2021-180)**

**General comments**

The experimental data on WDV in combined wave-current flows is most valuable for scientists and engineers working with nature-based coastal defences. The presented data is unique, relevant, complete, well-structured, and uses state-of-the art methodologies. The use of force transducers is viewed as an accurate way to measure forces on vegetation, which is not yet commonly used in other WDV experiments. However, several suggestions are made to improve the manuscript accompanying the data. I am confident that a revised manuscript will be an impactful article in ESSD.

Reply:

We thank the reviewer for the positive evaluation of our data and manuscript. We also very much appreciate the detailed suggestions provided by the reviewer, which help us improve the manuscript.

**Major specific comments**

Comment 1:

(Section 1) The perspective chosen in the introduction is in my opinion less suitable for ESSD. I would expect that the present dataset is presented from a background of earlier experimental datasets (Anderson & Smith, 2014; Augustin et al., 2009; Bradley & Houser, 2009; Foster-Martinez et al., 2018; Jadhav et al., 2013; Koftis et al., 2013; Mendez & Losada, 2004; Ozeren et al., 2014; Sánchez-González et al., 2011; van Veelen et al., 2020, and many others). The introduction insufficiently addresses why this new dataset is complementary to existing data sets. Instead, the authors approach the introduction from a perspective of why combined wave-current flows are relevant. While this is certainly important, this takes up a lot of space at the cost of relating the present work to prior experimental research.

Reply:

We agree with the reviewer on this comment. We revised the introduction to 1) include more existing experimental studies and 2) shorten the part related to combined wave-current flows and make it clear how this new dataset is complementary to existing datasets. These changes can be found in R60-79 and R148-149. A similar revision was also done in the abstract in R14-19.

In the introduction, we state that although there are several studies on waves with following currents, there are relatively fewer studies concerning opposing currents. The recent experiment of Maza et al. (2015) did include opposing currents, but they only considered submerged canopies. Emergent canopies were not tested. Additionally, all these experiment datasets are not openly assessable to the public. Thus, we present a

combined dataset composed of two flume experiments on WDV with underlying currents in both emergent and submerged conditions. Importantly, this combined dataset is now freely assessable to other researchers (Hu et al., 2020).

Comment 2:

(Section 2) Which type of vegetation is represented by the artificial vegetation used in the experiments? The authors refer mostly to the generic term of coastal vegetation. However, seagrass, salt marsh vegetation and mangroves have very different vegetation properties (dimensions, flexibility). Can the authors clarify how their artificial vegetation compares to real vegetation? And how useful are the outcomes for coastal vegetation types not represented by the artificial vegetation? If the authors choose provide clarifications in the manuscript, I recommend that they also do so in the abstract and section 1.

Reply:

We thank the reviewer for this comment. Indeed, we should clarify which type of vegetation is represented by the artificial mimics of our experiment. We used stiff mimics in the experiments to represent mangrove canopies. The obtained outcomes may be of use to the studies of flexible vegetation types. In fact, the $C_D$ relation derived in E14 has already been successfully applied in modeling wave dissipation by real flexible marsh plants, i.e., *S. Anglica*, *P. Maritima* and *E. Athericus* (van Veelen et al., 2021). Thus, we expect the outcome of the current study has a broader application range beyond the tested artificial vegetation.

In the revised MS, we have made the represented vegetation clear in the Abstract (R21, R29), Section 1 (R150) and the Discussion parts (R529) of the MS as suggested by the reviewer.

Comment 3:

(Section 2) The presented experiments are specifically designed to study wave-current interactions. Figure 1 shows that the in- and outlets are at a specific location in the water column (bottom/top). This raises the question on how the velocity profile is distributed over the water column. Can the authors comment on the velocity profile? Was this controlled? And how was this verified?

Reply:

We thank the reviewer for this comment. The velocity profiles over the water column (as shown in Figure 4) were measured at the middle of the flumes and far away from the in- and outlets. The vertical locations of the in- and outlets are expected to have an effect on the nearby velocity profiles but may not have an apparent impact on the velocity profiles in the canopies. In the experiment, the water depth and the discharge were controlled by the in- and outlets. The resulting velocity profiles were not controlled. The verification of the velocity profile is two-folded: 1) the depth-averaged velocity over the

presented velocity profiles is similar to the targeted flow velocity, especially in control cases without vegetation; 2) the presented velocity profiles are similar to previous experiments (e.g., Li and Yan, 2007; Pujol et al., 2013). To make these points clear, we revised the MS in R238 and R392 as:

"The velocity profiles were measured in the vegetation canopies, far away from both ends of the flume, to avoid the potential local influence of the in- and outlets. In each test, the water depth and discharge were set to the targeted values to create steady currents."

"The presented velocity profiles are similar to previous experiments (e.g., Li and Yan, 2007; Pujol et al., 2013)."

Comment 4:

(full manuscript) The E14 experiment has been previously published by the authors in Hu et al. (2014), hereafter Hu14. Compared to their previous paper, they have added an extra set of experiments (E19) and they have now published their data free to use. However, the partial reuse leads to several issues:

- The manuscript does not set out how the current manuscript should be read complementary to Hu14.

  Reply:

  We thank the reviewer for this comment, which reminds us to clearly state the complementary value of the current manuscript to H14. Now, we revised the MS to make this point clear by the two points listed below in R151-154 and R154-156:

  "In total, E14 conducted 314 tests, and E19 conducted 354 cases with different scenarios of incident waves, imposed current, vegetation density and submergence ratio (Table B1). E14 has systematically compared the variations of WDV and $C_D$ with or without co-existing following currents (Hu et al., 2014). As a complementary to the E14, E19 further conducted tests with opposing currents."

  "To our knowledge, it is the first freely assessable dataset that includes a wide range of current-wave combinations. Besides wave height variations, this new dataset contains detailed time series data of $F_D$ and $u$ in all the tests and velocity profiles in a few selected tests."

- The introduction reads "A subsequent laboratory study revealed that following current can either increase or decrease WDV (Hu et al. 2014)" (R48), followed by "contradicting conclusions" (R50, also R17) and "relevant datasets are still scarce" (R53). This suggests to the reader that you will address the conflicting views. However, the majority of the experiments presented (according to table B1), is identical to the ones that you seem to challenge in the introduction. It is no surprise that the results in this manuscript match the results in Hu14. I recommend that the

authors approach this manuscript as complementary to Hu14 rather than a test to of Hu14.

Reply:

We realize that mentioning the conflicting views of previous studies is not necessary, as our current manuscript will not address such conflict. Thus, the text related to the contradictory results has been deleted (R17 AND R50 of the original manuscript). Additionally, the revision has been made to clarify how the current manuscript can be complementary to Hu14, as mentioned in the reply above.

- The description refers to Hu14 to explain certain parts of the experimental setup (R104, R161, R167). I recommend that the experimental setup can be understood stand-alone from the present manuscript.

  Reply:

  We agree with the reviewer that the experimental setup should be understood stand-alone in the present manuscript. Thus, revision has been included to illustrate the experiment setup more clearly.

- As a positive comment, section 3 is concise and clearly shows the added value of E19 to E14.

  Reply:

  We thank the reviewer for this positive comment.

Comment 5:

(full manuscript) The referencing in this manuscript can be improved. While the selected literature is relevant, it could often be cited a more appropriate location. Some examples:

- R176: The work by Pujol et al. (2013) is very relevant for this manuscript, but why is it only mentioned at a trivial equation and not in section 3.2, which addresses a topic extensively covered by Pujol et al. (2013).

- R182: Why not Keulegan & Carpenter (1958) for the KC-number?

- R147,152: would it be possible to reference a more time-independent source? Also these references are not listed at the back of the manuscript.

- R378: This reference suggests that the conclusions of these experiments are made by the authors.

- R380: why only provide reference for root and leaf impacts, but not for flexible vegetation. There are many to choose from.

- The introduction would benefit from more references to relevant experimental studies (see also comment 1).

-

Reply:

We thank the reviewer for this detailed suggestion regarding references. All the points have been addressed in the revised manuscript. To ensure the referenced source can always be available to the readers (to be time-independent), we also include the documents of the applied instruments in the data repository, which are permanently assessable to the public with the data itself.

**Minor specific comments**

Comment 6:

(R17) "Previous studies" is rather vague. Could the authors be more specific on the type of studies?

Reply:

We agree with the reviewer on this comment. The related text has been deleted when we were addressing comment 4 above.

Comment 7:

(R58) "CD was introduced" feels off to me as it was already a common parameter in other scientific fields. I recommend rephrasing this sentence.

Reply:

We agree with the reviewer on this comment. The related text is revised as:

"$C_D$ is an empirical parameter that links known velocity ($u$, either from measurements or modeling) to the drag force exerted by vegetation stems ($F_D \sim C_D u^2$, Morison et al., 1950), which is directly related to WDV."

Comment 8:

(Figure 1)

- It might be helpful to mark FT2 as not working in the diagram.
- The green-red colouring may be an issue for those with colour blindness.
- The meaning of the asterisks is not immediately clear.
- Small z?

Reply:

We thank the reviewer for these suggestions regarding Figure 1. All the suggested comments are adapted in the revised Figure 1.

Comment 9:

(section 2.1)

Can the authors comment on how wave reflection was addressed? Especially for E14 this may be relevant as no wave absorber appears to be present.

Reply:

We thank the reviewer for this comment. To avoid the complexity induced by wave reflection, we only analyze the first 3-5 waves after the spinning up of the wave-makers. To make this point clear, we add the following text in R303-R304 as:

"To avoid the complex wave reflection conditions, we only analyzed the first 3-5 waves after the spinning up waves. We turned off the wave-makers after about 20 waves in each test."

Comment 10:

(R144) The water depth in E19 is shallow. Were the authors able to maintain regular waves under these conditions? The velocity time series in Figure 5 appears to contain some nonlinear components. If not, what would be the implications? And was any additional impact of bottom friction observed?

Reply:

We thank the reviewer for this comment. Indeed, there can is nonlinearity in the created waves in both E14 and E19. This nonlinearity leads to weak recirculation in the flume, which can be observed from the negative time-mean current velocity in pure wave cases (Figure 4). This recirculation in the flumes is common in wave flumes and has been attributed to Stokes drift (Hudspeth & Sulisz, 1991; Hu et al., 2014). The effect of this nonlinearity and recirculation on WDV has been discussed in Hu et al. (2014). Additionally, this recirculation can also occur in field conditions as wetlands are often bounded by landward dikes. These dikes are closed boundaries similar to the baffle plates in confined flumes, which can also induce Stokes drifts. The impact of bottom and sidewall friction can be observed in control tests without vegetation (VD0) and documented in the dataset.

To make this point clear, we add the above text in the second paragraph of section 2.3 wave conditions in E14 and E19.

Comment 11:

(R156)

I recommend devoting a separate section to this paragraph, e.g. 2.3 wave conditions.

Reply:

We thank the reviewer for this suggestion. We have created a separate section to this paragraph as "2.3 wave conditions".

Comment 12:

(R166) The middle water depth is chosen as representative for the depth-averaged velocity. However, Figure 4 shows that velocities are attenuated inside the canopy and amplified above the canopy. These appear to be two separate regimes that cannot easily be captured at a fixed water depth. The main use for u is the derivation of the drag force. With this in mind, I would think that a measurement that is representative for the canopy regime is most appropriate. Have the authors considered using a velocity halfway the vegetation height rather than water depth? It seems that the current definition would not be sufficient if the canopy height is below half water depth.

Reply:

We agree with the reviewer on this comment. Indeed, the velocity measured at the halfway of the vegetation height is a more universal option for various tests with different water depths. However, as in our experiments, the tested vegetation submerged depth is small, $u$ at half water depth and at half vegetation height is similar. This can be observed from the velocity profile measurements in Figure 4, which shows that the velocity inside the canopy is almost uniform in both experiments. Thus, in our experiments, $u$ at half water depth is still suitable. To make this point clear, we add the following text in R262-265 as:

"Since the velocity measurement was to obtain representative in-canopy velocities, thus in submerged canopies, it was perhaps more suitable to measure velocity at half of the canopy height rather than at half water depth. However, given the relatively shallow water depths tested in both E14 and E19, velocities obtained at both positions were similar, as shown in the vertical velocity profiles (see Figure 4)."

Comment 13:

(Eq 4,5) Do umax and umin include the contribution of the mean current?

Reply:

$u_{max}$ and $u_{min}$ do include the contribution of the mean currents. They are the measured peak flow velocities in the positive and negative directions in a wave period, which are

influenced by the mean currents. To make this point clear, we add the following text in R336 as:

"Both $u_{max}$ and $u_{min}$ change with co-existing mean currents."

Comment 14:

(Eq 5) The KC-number is based on measured velocity. How can the velocity be estimated when no measurements are available? It will be interesting to shortly address this in Section 4.1, where potential applications are discussed.

Reply:

We thank the reviewer for this comment. When no measurements are available, velocities in KC-numbers can be estimated by using linear wave theory. In numerical modeling studies, we can also first assume Cd=1 to obtain a first estimation of the velocity in KC-numbers, and then move towards more accurate Cd and KC-numbers by iterations. To make this point clear, we add the following text in Section 4.1 in R530-532 as:

"When velocities are unknown to define *KC* numbers, the velocities may be estimated by linear wave theory or by numerical iterations. For the latter case, an initial $C_D$ value can be set as 1 to start the iteration."

Comment 15:

(R195) Did you find any difference in turbulence between the two canopy types? Or was this not measured?

Reply:

We thank the reviewer for this question. Turbulence was not measured in our experiments as the main focuses of the current studies are WDV and Cd variations.

Comment 16:

(Figure 2)

It is confusing to me why subplots a-c are fitted but d-f are not. I recommend making this consistent within the figure, and within the methodology as a whole.

Reply:

We followed the reviewer's suggestion and include fitted lines in all the subplots in the Figure 2.

Comment 17:

(Figures 2,3,6,C1) A confidence interval and/or a measure of goodness-of-fit would be valuable. This may also concern individual measurement points where relevant. I noticed that you already included goodness-of-fit measures in your published database. However, I believe that it will also benefit readers to get an impression of the variability of the data from the manuscript.

Reply:

We thank the reviewer for this comment. $R^2$ values were introduced as a measure of goodness-of-fit in Figure 2, 6 and C1. Figure 3 only contains data points. Thus, goodness-of-fit measures are not included in Figure 3.

Comment 18:

(Section 3.2) While the velocity and force data are part of the database, section 3.2 feels out of place in its current form. This section is not introduced anywhere before in the manuscript. Those focus fully on WDV and CD. I recommend that this is addressed.

Reply:

We agree with the reviewer on this comment. We have included the following text at the end of the Introduction to justify the inclusion of velocity and force measurements in R155-156:

"Besides wave height variations, this new dataset contains detailed time series data of $F_{D}$ and $u$ in all the tests and velocity profiles in a few selected tests, as these data are important in assessing $C_D$ and WDV."

In the first paragraph of section 3.2, we add the following text as a connection to WDV as:

"Since the variation of WDV in different flow conditions is closely related to the spatial velocity structures, we measured the vertical velocity profiles in a few tests with the same wave condition but different accompany currents (Figure 4)."

In the second paragraph of section 3.2, we revised the text as following to make the text linked to $C_D$:

"Apart from the vertical velocity structures, we also include the raw data of the temporal variations of velocity ($u$) and the acting force ($F$) on vegetation mimics at multiple locations along vegetation canopies to derive $C_D$ for all the tested cases (Figure 5)."

Comment 19:

(Section 3.2, Table B1). It is unclear during which tests the a full velocity profile was measured. As far as I understand, this was only done for a subset of the experiments. Could you please specify which tests had the full velocity profile measured?

Reply:

We agree with the review on this comment. We have marked the cases with velocity profile measurements in the Table B1.

Comment 20:

(Figure 4)

- I recommend leaving the no veg and current-veg cases out for clarity.

Reply:

We thank the reviewer for this comment but leaving out no veg and current-veg cases would not reveal the effect of vegetation on the velocity structure. We decided to include a larger figure in the revised manuscript to improve the clarity.

- Please add the meaning of each dashed line to the caption.

Reply:

We thank the reviewer for this comment. The meaning of the dashed lines is added in the caption of Figure 4.

Comment 21:

(Figure 5)

Could the authors clarify in the caption which timeseries are measured and which are calculated?

Reply:

We thank the reviewer for this comment, and the caption is revised as suggested.

Comment 22:

(R361, R380-386) The CD relation of E14 was also applied in van Veelen et al. (2021, Coast. Eng.), which is a use case of expanding the CD-relation such that it is useful for flexible vegetation too. This may be relevant, especially as the authors discuss expansions towards more realistic vegetation conditions in R380-R386.

Reply:

We thank the reviewer for this comment. The reference (van Veelen et al. 2021, Coast. Eng.) has been added in this part of the discussion, and following text has been added in the discussion in R565-566:

"In fact, the $C_D$ relation derived in E14 has already been successfully applied in the modelling of wave dissipation in real flexible marsh plants, i.e., *S. Anglica*, *P. Maritima* and *E. Athericus* (van Veelen et al., 2021)."

Comment 23:

(Section 4.2) The title of section 4.2 does not reflect the contents well in my opinion. As a result, the main message of this section remains unclear to me.

Reply:

We thank the reviewer for this comment. We have changed the title of this section to "A unique dataset for further researches in WDV". The first paragraph summarizes the observed rich variations in WDV from our dataset. The second paragraph discusses the limitation of the current dataset (e.g., excluding vegetation flexibility) and possible applications in the future, e.g., in theoretical and numerical models. Thus, we think this title covers the contents of these two paragraphs.

**Technical corrections**

- R10: is the institute of reference 5 correct?
- R30: systems has
- R161: possible influenced
- R190: Hpw and Hcw is
- R197: the accompany currents
- R364: lastly (capitalization)
- R372: in adequate

Reply:

We thank the reviewer for these comments. We confirm that the institute of reference 5 is correct. All the other suggestions from the reviewer have been addressed in the revised manuscript.

---

## Author Comment (AC2)

**Reply to RC2 regarding manuscript:**

**Laboratory data on wave propagation through vegetation with following and opposing currents (essd-2021-180)**

**General comments**

This manuscript provides a data set for two flume experiments that investigate the impacts of vegetation on wave dissipation in the presence of both current and wave. The data set is rich and should be valuable for future investigation of the impacts of vegetation on wave dynamics in coastal areas. The scientific contribution of the manuscript is weak because the difference between this study and previous studies is not clear. Consider that this is a data description paper, I think the paper may be a good fit after revision.

Reply:

We thank the reviewer for the overall positive evaluation of our dataset and the note on the scientific contribution of the manuscript. For the latter, we made the following action to clarify the contribution of the manuscript:

In the introduction, we state that although there are several studies on waves with following currents, there are relatively fewer studies concerning opposing currents. The recent experiment of Maza et al. (2015) did include opposing currents, but they only considered submerged canopies. Emergent canopies were not tested. Additionally, all these experiment datasets are not openly assessable to the public. Thus, we present a combined dataset composed of two flume experiments on WDV with underlying currents in both emergent and submerged conditions. Importantly, this combined dataset is now freely assessable to other researchers (Hu et al., 2020).

**specific comments**

I think the organization of the manuscript needs improvement. The manuscript mentions that the data sets include 668 tests. What are the conditions of each test? I think there should be a table on the conditions of each test and the measurements being made for each test as a summary of the data set? The conditions of the vegetation, e.g. modulus, size, submergence ratio, is not clear to find, at least for me. I think this information should also be summarized and the test conditions should be organized by categories.

Reply:

We thank the reviewer for these comments, which mainly concerning the clarity of the experiments. The conditions of each test are included in a summarizing table, i.e., Table B1. The vegetation conditions are included in the first paragraph of Section 2.1 and Section 2.2 for the two experiments, respectively. Furthermore, the tested wave conditions are now detailed as a separated section, i.e., Section 2.3, as suggested by the reviewer RC1.